# From Wuhan to COVID-19 Pandemic: An Up-to-Date Review of Its Pathogenesis, Potential Therapeutics, and Recent Advances

**DOI:** 10.3390/microorganisms8060850

**Published:** 2020-06-04

**Authors:** Ikrame Zeouk, Khadija Bekhti, Jacob Lorenzo-Morales

**Affiliations:** 1Instituto Universitario De Enfermedades Tropicales y Salud Pública de Canarias, Universidad de La Laguna, Avda. Astrofísico Fco. Sánchez, S/N, La Laguna, Tenerife, 38203 Islas Canarias, Spain; 2Faculty of Sciences and Techniques, Sidi Mohamed Ben Abdellah University, PB 2202, Fez 30000, Morocco; bekhti.bki4@gmail.com; 3Departamento de Obstetricia, Ginecología, Pediatría, Medicina Preventiva y Salud Pública, Toxicología, Medicina Legal y Forense y Parasitología, Universidad De La Laguna, La Laguna, Tenerife, 38203 Islas Canarias, Spain

**Keywords:** COVID-19, SARS-CoV-2, pathogenicity, therapeutic advances, prevention

## Abstract

The emergence of a novel human coronavirus (SARS-CoV-2) causing severe contagious respiratory tract infections presents a serious threat to public health worldwide. To date, there are no specific antiviral agents available for this disease, currently known as COVID-19. Therefore, genomic sequencing and therapeutic clinical trials are being conducted to develop effective antiviral agents. Several reports have investigated FDA-approved drugs as well as in silico virtual screening approaches such as molecular docking and modeling to find novel antiviral agents. Until now, antiparasitic drugs such as chloroquine have shown the most relevant results. Furthermore, there is an urgent need to understand the pathogenesis of this novel coronavirus, its transmission routes, surface survival and evolution in the environment. So far, the scientific community has indicated a possible transmission of COVID-19 via blood transfusion which is challenging in the case of asymptomatic individuals. Protocols for pathogen inactivation are also needed. In this paper, we reviewed recent findings about this life-threatening pandemic.

## 1. Introduction

Belonging to coronaviridae family, coronaviruses (CoVs) are enveloped non-segmented positive-sense RNA viruses widespread in humans and animals [1]. CoVs were identified as causative agents of two previous epidemies, SARS (Severe Acute Respiratory Syndrome) and MERS (Middle-East Respiratory Syndrome) that both had previously negative economic and social impacts [2,3]. In December 2019, a new infectious respiratory disease caused by a novel CoV emerged in Wuhan, Hubei province, China [4]. High-throughput sequencing technology provided new insights into the identification of this virus [5,6] which was temporally named 2019-nCoV (2019 novel coronavirus) and closely related to SARS virus [7], then designated as SARS-CoV-2 [8]. The epidemic scale of SARS-CoV-2 infection has been continuously increasing around the world with more than 6,057,853 confirmed cases and 371,166 deaths [9]. Currently, there is no specific vaccine against SARS-CoV-2 infection which is critical. Therefore, scientific and social measurements are urgently needed for an effective pandemic containment. Several pharmacological and non-pharmacological approaches have been used worldwide with various clinical efficacy and outcomes. The aim of this review is to summarize recent advances on SARS-CoV-2 pathogenicity, its mechanism of cell entry, transmission mode and surface adhesion.

In addition, we also discuss in vitro assays for the discovery of effective antiviral drugs as well as other therapeutic approaches to control this emerging pandemic.

## 2. SARS-CoV, MERS-CoV and SARS-CoV-2: Is There a Link?

CoVs have developed three groups, classified serologically depending on the host range and genome sequence [10]. The human described CoVs are HCoV-229E and HCoV-OC43 identified in mid-1960 and associated with common colds [11], SARS-CoV which is the most pathogenic strain responsible for life-threatening pneumonia in 2002 [12], HCoV-NL63 in 2004, HCoV-HKU1 in 2005 [13] and MERS-CoV in mid-2012 [14]. The evolution of these viruses occurred via some features such as genome flexibility. Several authors have sequenced the genome of the novel SARS-CoV-2, and compared itwith that of previous CoVs. Zhou et al. have described the full-length genome sequences obtained from infected patients, and they have detected similarities between the novel virus, bats virus and SARS-CoV. The sequences were identical to those of SARS-CoV (79.6%) with some changes in four out of five of the key residues in the receptor-binding, and 96% to that of bats with some differences in the three short insertions in the N- terminal domain of S gene sequences concluding that primers could differentiate SARS-CoV-2 from the other human CoVs [15]. Although the sequence of 3CLpro (3-chymotrypsin-like protease) protein of SARS-CoV-2 has exhibited strong similarities to bat SARS-like CoVs (99.02%), SARS-CoV (96.08%), then MERS-CoV (87%), 12-point mutations have been noted namely Val35Thr, Ser46Ala, Asn65Ser, Val86Leu, Lys88Arg, Ala94Ser, Phe134His, Asn180Lys, Val202Leu, Ser267Ala, Ser284Ala and Leu286Ala [16]. In addition, a peculiar furin-like cleavage site in the spike protein has been identified in SARS-CoV-2 and not in other SARS-like CoVs. Obviously, this cleavage site could be involved in transmissibility and pathogenesis [17].

Furthermore, SARS-CoV-2 is able to multiply better in primary human airway epithelial cells than in standard tissue, contrary to SARS-CoV and MERS-CoV that infect intrapulmonary epithelial cells more than cells of the upper airways [18,19]. Since SARS-CoV is known by a remarkable adaptation and a high mutational profile, it was proposed that SARS-CoV-2 will behave more like SARS-CoV than MERS-CoV.

## 3. Entry Events and Pathogenicity

### 3.1. Description

The pathogenicity of SARS-CoV-2 infection is mainly associated to its structural features (depicted in Figure 1). The SARS-CoV-2 structure is similar to that of other CoVs which are composed by spike (S) protein, a trimeric glycoprotein with two functional domains namely S1 and S2 that play a crucial role in host cell entry. S1 initiates the viral entry via the receptor binding domain, while S2 is involved in the induction of fusion between cell and viral membranes during endocytosis. S2 contains amino acid sequences required for viral infectivity. Thus, the induction of fusion involves cleavage of S proteins by proteases present in the host cells. The second viral structural proteins designated membrane (M) protein, which is the most abundant and inserted in the viral particle, involved in the maturity and shape of the virion. The envelope (E) is the third protein on the surface and is a non-glycosylated transmembrane protein found in small quantities that facilitates the assembly and the budding process. In addition to internal nucleocapsid (N) proteins, which are the most conserved among the other proteins of CoVs, and contains two domains, a N-terminal domain and a C-terminal domain, able to bind RNA and leading to its incorporation into the virion. This protein is necessary for the RNA genome encapsulation and replication [20,21].

Besides the four main structural proteins, there are sixteen nonstructural proteins (nsp1–16). Several of them direct the viral replication-transcription machinery in CoVs [22], while some of them display other functions. Nsp1, the N-terminal protein coded by the gene 1, inhibits the host gene expression by binding and inactivating the translation activity of the 40S ribosomal subunits [23]. While nsp2 is involved in the disruption of intracellular host signaling [24]. For nsp3, it is the largest protein encoded by CoV genome, and it plays many roles in the viral life cycle such as binding other viral nsps and inducing modifications in the host proteins to antagonize the host innate immune response [25]. Indeed, an earlier paper has analyzed the Open Reading Frame 1ab (ORF1ab) of SARS-CoV-2 and found that positive selective pression and mutations in nsp2 and nsp3 could differentiate COVID-19 pathogenesis from SARS [26]. As for the transmembrane nsp4, it is a critical component for assembly of the replication complex on double membrane vesicles [27]. In addition, the highly conserved nsp5 is the viral main protease, which contains a chymotrypsin-like fold that cleaves the nsp4–nsp11 and nsp4–nsp16 necessary for the virus replication [28]. Nsp6 is a multiple-spanning transmembrane protein able to limit the expansion of authophagosomes and omegasomes [29]. Regarding nsp7, it constitutes a component role player of SARS-CoV replicase polyprotein, while nsp8 is involved in initiating the RNA synthesis. This protein exhibits noncanonical template-dependent polymerase and adenylyltransferase activities [30]. The other nsps appear to be involved in different viral functions such as RNA binding (nsp9) [31], activation of multiple replicative enzymes (nsp10 and nsp13) [32,33], endoribonuclease activity (nsp11 and nsp15) [34,35], RNA polymerase activity (nsp12) [36], modulation of the innate immune response (nsp14) [37], and 2′O-MTase activity (nsp16) [38].

Information on the specific interactions among nsps is also emerging. For example, nsp8 is known by its ability to interact with nsp7 and the functional open reading frames (ORFs) accessory protein. The interaction between SARS-CoV ORF6 and nsp8 has been confirmed based on different assays, including yeast-2-hybrid and coimmunoprecipitation experiments indicating that interactions among these proteins contribute to the SARS replication [39]. Furthermore, SARS-CoV nsp7 found to enhance RNA binding by nsp8, while the complex has primer extension RNA polymerase activity [40]. Recently, a comparative phylogenetic analysis between SARS-CoV-2 and SARS-CoV revealed that nsp7b and nsp8 have been conserved among SARS-CoV-2 and are associated with the evolution of COVID-19 [41]. The complex nsp3,4,6 has been also studied, showing that these three proteins might bind together to support the rearrangement of membrane required for the genome replication [42].

### 3.2. Cell Entry and Pathogenicity: ACE2 and TMPRSS2

The cell entry stage is critical for the induction of viral infection. The entry events of an infection are initiated by the viral S-protein binding to the host cell receptors. Various entry routes have been described in CoVs. However, only SARS-CoV engages the angiotensin-converting enzyme 2 (ACE2) as entry receptor [43]. Reported similarities between SARS-CoV and SARS-CoV-2 allowed researchers to explore genomic investing ations to verify if SARS-CoV-2 engages the same receptor to enter the cell. Zhou et al. have developed a qPCR-based detection method based on the receptor-binding domain (RBD) of S gene since it is the most variable region of the genome, indicating that SARS-CoV-2 seems to have an RBD that binds with high affinity to ACE2 [44]. In the same context, the analysis of RBD domain of SARS-CoV-2 S-protein has shown that the S-protein–ACE2 binding pathway is involved in the health damage during infection [45]. Furthermore, the ACE2-binding ridge in SARS-CoV-2 RBD has a more compact conformation in comparison to that of SARS-CoV RBD which increases its ACE2-binding affinity [46]. Moreover, engagement of this same receptor has been confirmed by a replication-defective vesicular stomatitis virus (VSV), and it was suggested that SARS-CoV-2 could use the cellular serine protease for S-protein priming (TMPRSS2) involved in the disease spread within the host cells [47]. In fact, the co-expression of ACE2 and TMRSS2 has been highly described in nasal epithelial cells highlighting a critical role in viral infection, spread and transmission [48]. This is strong evidence that the ACE2 is the key receptor for the spike glycoprotein of SARS-CoV-2. These findings have been widely confirmed in literature [49,50]. Understanding this type of information is useful for the development of antivirals able to inhibit the virus-cell attachment and fusion.

### 3.3. Endocytic Pathways

To infect the host cell, various entry routes have been described in CoVs. However, the endocytic pathway is the most engaged in CoVs viral entry. For instance, the key mechanism for viral entry of SARS-CoV is clathrin-dependent as well as clathrin- and caveolae-independent via pH- and receptor- dependent endocytosis [51]. Taking into consideration the similarities between SARS-CoV and SARS-CoV-2 receptor choice, it was suggested that SARS-CoV-2 may involve the same endocytic pathway, especially after showing its sensitivity towards the lysosomotropic chloroquine [52]. Targeting the endocytic route may provide a novel therapeutic strategy.

## 4. COVID-19 Symptoms

In addition to the most common clinical manifestations of COVID-19 (fever, dry cough and dyspnea), recent studies have revealed that gastrointestinal symptoms are not negligible, and clinicians should be aware since the incidence of diarrhea, nausea, vomiting and abdominal discomfort has been confirmed [53], in addition to cutaneous manifestations and complications [54,55]. Moreover, olfactory dysfunction such as the loss of smell and taste has been also described in patients with COVID-19 [56,57]. This dysfunction could be linked to encephalitis or cerebropathia symptoms that may reveal the disease [58]. An early diagnosis of COVID-19 may lead to a perception of improved survival and reduce the risk of medical staff infection.

## 5. Transmission Routes

Recent reviews have reported that SARS-CoV-2 is spreading rapidly through different conditions [59], here we discuss the most common way which is transmission and close contact. Person-to-person transmission has been reported as a critical route for COVID-19 spread, taking into consideration the widespread transmission in China and around the world. The virus is spread via direct contact or in small contaminated droplets coughed or sneezed into the air by an infected person. Indeed, a higher rate of SARS-CoV-2 has been detected in saliva than other respiratory samples [60]. To et al. have confirmed this finding and showed that SARS-CoV-2 was highly observed in the self-collected saliva [61]. Moreover, in an observational cohort study in Hong Kong, authors have revealed that salivary viral load was highest during the first week after symptom onset, and the viral RNA still detected in samples of posterior oropharyngeal saliva for 20 days or more [62]. Unfortunately, recuperative patients can act as reservoir for the transmitted infection.

Another critical point in transmission features, is the pre-symptomatic spread. In a case report, the timeline story of one isolated asymptomatic patient has been described. First, the patient has came from a focus of COVID-19, but had registered normal Chest CT images, normal C-reactive protein level, normal lymphocyte count, and negative RT-PCR. After few days, the RT-PCR of this patient was positive. Nevertheless, the relatives of the described patient were all RT-PCR positive before her which allowed researchers to suggest an asymptomatic transmission [63]. Unfortunately, this hypothesis was confirmed in Singapore and Tianjin; the viral infection was transmitted on average 2.55 and 2.89 days before symptom onset [64]. Likewise, SARS-CoV-2 has been detected in asymptomatic individuals which constitutes a central risk of COVID-19 spread [65]. In addition, the human-human transmission within families also occurs; in China the viral RNA has been detected in patients’ feces performing a RT-PCR analysis [66]. Providing information on the transmission routes is very important for an active surveillance, especially in the case of a newly emerged pandemic.

### Adopted Measures in Reducing COVID-19 Transmission

Since it is supposed to be efficient in the local control of a contagious pandemic, strategies and protocols have been developed to decrease the transmission, such as isolation of cases and lock-down contacts. The efficacy of this measure was confirmed by developing a stochastic transmission model parameterized to the COVID-19 disease [67]. A typical case in public transport has been described, a patient with COVID-19 wore a face mask and did not contaminate persons that were also wearing masks in the same minibus, however, in another vehicle in the absence of masks, individuals were infected [68]. This is in line with previous findings highlighting that masks help limit the spread of disease and decrease the detection of viral RNA [69] especially when the transmission via aerosols is also possible [70].

Furthermore, COVID-19 transmission via transfusion is also plausible and staff in blood centers and banks should follow several measurements such as donor’s inspection, RNA, virus-related antibody screening and inactivation of CoVs in blood products [71].

## 6. SARS-CoV-2, Surfaces Persistence, Hospital Environment and Disinfection

The persistence of CoVs on animate and inanimate surfaces has been described, and correlation between virus persistence and parameters such as temperature, relative humidity, viral load and viral strain has been analyzed. The virus can persist on different materials like steel, metal, wood, plastic, glass, paper, surgical glove and others from 2 h, 9 days and more than 28 days at 4 °C. Moreover, the virus “prefers” high relative humidity [72]. The use of disinfectant such as 62–71% ethanol, 0.5% hydrogen peroxide and 0.1% sodium hypochlorite has shown important results in reducing the infectivity of the virus in a short time (1 min) [73]. In the hospital, the transfer of infection by an intermediary air, surface environment and personal protective equipment has been examined performing a specific RT-PCR that targets the RNA-dependent RNA polymerase and E gene of SARS-CoV-2, results have shown a high environmental contamination caused by infected patient via respiratory droplets and feces, and suggest that the virus in small droplets can be transmitted by airflows and persist on equipment [74]. Likewise, SARS-CoV-2 comparably to SARS-CoV-1 can stay viable and vital in aerosols for hours and on surfaces for days [75]. Taken together, these findings, environmental and hand hygiene are necessary to limit and control the transmission and spread of COVID-19.

## 7. Could Molecules Used against SARS-CoV Be a Source to Eradicate SARS-CoV-2?

Resemblances between SARS-CoV and SARS-CoV-2, especially the same affinity to ACE2 receptor, could suggest that efficient molecules used against SARS-CoV may provide valuable information to develop effective anti SARS-CoV-2 drugs. Certainly, there are numerous investigations evaluating the in vitro potential of drugs and compounds against SARS-CoV, but here we report the most tested drugs. For instance, it is worth noting that ribavirin—a purine nucleoside analogue and licensed broad-spectrum antiviral [76]—was widely tested against SARS-CoV showing good activity by inhibiting the virus replication [77,78,79]. In addition, remdesivir has been also reported as efficient with an EC_50_ of 0.069 µM and selectivity index SI > 144.93 [80,81]. It is a monophosphoramidate prodrug and an adenosine analogue used as antiviral agent [82].

## 8. Current in Vitro Tested Drugs against SARS-CoV-2, Chloroquine and Its Mechanism of Action

The discovery of new drugs needs a long time, which is a real obstacle in the case of pandemic. Subsequently, several authors have evaluated the antiviral efficacy of known drugs (Table 1).

Among the tested drugs, the antimalarial chloroquine and its derivatives were the most reported as potential treatment against SARS-CoV-2. Twenty FDA approved drugs have been screened against SARS-CoV-2 since the safety profile and benefits of these drugs are known. Seventeen of these compounds were active against SARS-CoV-2 strain with an IC_50_ < 50 µM at non-cytotoxic concentrations. Next, the mechanism of action of chloroquine (CQ) and hydroxychloroquine (HCQ) as the promising treatment has been elucidated, highlighting their inhibitory potential via cytopathic effect and reduction in viral mRNA level, then replication [91]. In a comparative in vitro assay between HCQ and CQ efficacy at different multiplicity of infection (MOI), HCQ was less potent than CQ. This result was confirmed by variations in the expression of nucleoprotein levels of the virus using immunofluorescence microscopy. Besides, the time of addition experiments has been performed and revealed that both compounds displayed inhibitory effect in the entry and post entry stage, in addition to morphological and size modifications in early endosomes and endolysosomes [89]. Furthermore, the interesting in vitro activity of QC has been demonstrated in addition to remdesivir previously used as antiviral agent. In the case of SARS-CoV-2, remdesivir seems to act at a stage post virus entry [87]. However, in another investigation, it was reported that HCQ showed more efficacy than CQ [88]. Several factors and parameters could explain these slight differences, such as the virus culture adaptation in vitro and the use of different ranges of MOI. Nevertheless, the potentiality of these drugs cannot be denied.

The clinical administration of remdesivir was established in a case report for the first patient infected by SARS-CoV-2 in United States. However, the administration decision was linked to the high gravity of the patient clinical status [92].

The common efficacy of chloroquine and its derivatives has been discussed, highlighting a possible mechanism of action through an increase of the pH of intracellular vesicles which influences the functionality of the virus, since it needs acidification of endosomes. Another mechanism was determined, which is the induction of altered ACE2 glycosylation that block the S- protein binding [93]. In more detail, a recent study has elucidated a new mechanism of action of CQ and HQC performing a modeling method, a new type of ganglioside binding domain has been described at the tip of N-terminal domain of SARS-CoV-2 S-protein. This ganglioside was involved in the virus-lipid rafts attachment facilitating the contact with ACE2 in the host cell membrane. Furthermore, the viral S-protein was unable to bind these gangliosides in the presence of CQ and HQC [94]. In addition, it was reported that CQ inhibits endocytosis nanoparticles and reduces the expression of phosphatidylinositol binding clathrin assembly protein (PICALM) which an adaptor responsible of sensing and driving the membrane curvature [95]. CQ was recommended to be added to the list of trial drugs in the sixth edition of the Guidelines for the Diagnosis and Treatment of COVID-19 published by National Health Commission of the People’s Republic of China [96] and many countries start its clinical administration.

In addition to these drugs family, ivermectin—an antiparasitic drug, also beneficial in treatment of viral infections—has revealed an in vitro promising inhibitory effect, and a remarkable reduction in viral RNA with 93% in 24 h. Interestingly, a treatment for 48 h has shown around 5000-fold reduction (99.8%) in comparison to the control [83].

Recently, Choy et al. have evaluated some compounds from different origins and found that some of them displayed efficient potential via inhibition of replication, indicating a synergistic effect between remdesivir and emetine at IC_50_ lower (6.25 and 0.195 µM respectively) than those screened separately (Table 1). Unfortunately, in the same study, ribavirin which is undergoing clinical trials, found to be non-effective against SARS-CoV-2 [84].

Taking the reported data together, more studies are needed in the case of in vitro assays, results should be confirmed and completed by in vivo tests and preclinical studies before recommending the human administration, even if the drugs are known, since the use against SARS-CoV-2 is not yet approved. Moreover, it is important to note that antiparasitic drugs presented the most relevant results, drugs such as chloroquine, ivermectin and emitine have shown the best effects leading us to propose a hypothesis “Could antiparasitic drugs be a promising source against SARS-CoV-2?”

## 9. Recent Advances in Therapeutic Alternatives against SARS-CoV-2: Virtual Screening, Neutralizing Antibodies and Others

New approaches for therapeutic alternatives other than testing the activity of known drugs in vitro have been developed. Scientists have moved to develop vaccines targeting different viral components, the mechanism of cell entry and the host response. The concept of neutralizing antibodies was widely reported, the S- protein was targeted since it is mandatory for the vital infection. Single-domain antibodies termed “VHHs” (Heavy-chain antibodies) have been isolated from a camelid “llama” immunized with prefusion- stabilized spikes of CoV, showing a cross-reactivity between SARS-CoV-2 S- and SARS-CoV-1 S-directed single-domain antibodies. These VHHs were able to neutralize SARS-CoV-2 S pseudoviruses as bivalent human IgG Fc-fusion [97]. In another study, a machine learning model has predicted the antibody response for COVID-19 via a screening of 1933 virus antibody sequences collected from a dataset, and contains antibody-antigen sequences of various viruses such as SARS, Ebola and Influenza with the IC_50_ values of neutralization panel, the inhibitory effect of 8 mutant stable antibodies has been noticed against COVID-19 at a class prediction of 100% showing that a mutation in amino acids (from Methionine to Tyrosine) is beneficial and increases the antibodies affinity against COVID-19 [98]. In the same vein of virtual prediction and repurposing methods, a wide range of investigations have been reported, using different statistical modeling of drugs and compounds from various libraries such as medicinal plants [99], FDA-approved drugs [100], antivirals drugs [101] against different targets including nucleotide inhibitors [102], protease and RNA polymerase [103], viral spike protein [104].

The use of virtual screening (VS) in therapeutics is not new, applications have been widely described [105] and showed promising results. The VS could save time, direct research and predict the efficacy and profile safety of samples. However, VS has some limitations depending on the used method, target, interactions and applications especially when the physicochemical factors are not completely understood [106].

Recently, a new antiviral approach represented by PAC-MAN (prophylactic antiviral CRISPR in human cells) strategy has shown relevant results against SARS-CoV-2. Abbot et al. have demonstrated that CRISPR-Cas13d inhibited the viral function and replication by targeting and cleaving all viral positive-sense RNA. Using bioinformatic analysis, a group of only six crRNAs was demonstrated to target a wide range of CoVs [107]. Nevertheless, the host viral response is also critical, especially the immune mechanisms that are necessary to eliminate the virus or halt its growth. It was reported that most patients with severe COVID-19 present high serum levels of pro-inflammatory cytokines which may cause shock and tissue damage in different organs such as liver, kidney and heart, in addition to respiratory failure [108]. Furthermore, severe COVID-19 patients displayed an immune dysregulation and macrophage activation syndrome. However, the interleukin-6 blocker Tocilizumab (TCZ) has shown relevant efficacy by avoiding the immune dysregulation in vitro and in patients [109]. While TCZ administration did not reduce ICU admission nor mortality rate, suggesting that more investigations about the TCZ efficacy are still needed [110].

## 10. Latest Prophylaxis and Management Strategies for COVID-19

Rigorous non-pharmaceutical measures should be implemented for an effective containment of COVID-19. Focusing on the pandemic dynamics and epidemiology is crucial and may provide helpful data. SARS-CoV-2 has been detected in a patient suffering from influenza-like illness in Wuhan suggesting an early community transmission in January and suggesting the urgent need for additional serological examination among populations to gain information about the history of the pandemic [111]. Serological assays play an important role in the detection of a convalescent case and transmission links. The generation of epidemiological data in more details and update is mandatory and could be helpful in pandemic outbreaks [112] and give an idea about geographical distribution and cases evolution. A mathematical modeling of geographical spread has been performed using a network-based approach to predict the COVID-19 risk at different geographical locations and future hotspots. The model has provided pertinent information on the spread based on the spatial distribution of existing cases, human mobility patterns, and administrative decisions which is an effective method for broader geographical countries [113].

Among other efficient strategies, social measures and safety practices could minimize the spread of this pandemic. Wearing face masks, it is considered as altruism and solidarity [114] especially with the revelation of asymptomatic transmission and stability of the virus in aerosols. A conceptual model has been developed using the suppression strategies based on social distancing and active protection and quarantining. This model concludes that lock-down is very effective for pandemic suppression [115]. The important role of safe behavior and interventions in the reduction of COVID-19 transmission has been confirmed [116]. Interestingly, the quantification of SARS-CoV-2 transmission may serve as an epidemic control, performing a contact-tracing App which builds a memory of proximity contacts and provide notifications about contacts of positive cases [117].

In another aspect of disease prevention, CQ and CQH have been recommended as prophylactic agents which is an important preventive initiative [118]. Providing this type of information could orient scientific committee and enhance the undertaken strategies to manage, control and limit at least the spread of COVID-19.

## 11. Conclusions

The COVID-19 pandemic caused by SARS-CoV-2 has affected communities around the world and caused devastating consequences. The scientific response to the COVID-19 outbreak is rapid, ranges from genomic investigations to drugs design and discovery. Based on data summarized in the present review, it is important to note that FDA approved drugs are most commonly tested since their efficacy and safety profile is known, which is helpful especially in pandemic situations. From results reported by different authors, we suggest some points that could be helpful in treatment option, such as the search for drugs from natural products known by their ability to inhibit the serine protease (TMPRSS2) priming and then S-protein binding, to induce neutralizing antibodies and/or to destroy the interaction between the receptor binding domain of SARS-CoV-2 and ACE2. Moreover, the use of drug combinations may serve as an approach for therapeutic interventions against the new emerging SARS-CoV-2. Interestingly, the consideration of hygiene conditions and contact isolation could play an important role in limiting the spread of COVID-19.

## Figures and Tables

**Figure 1 microorganisms-08-00850-f001:**
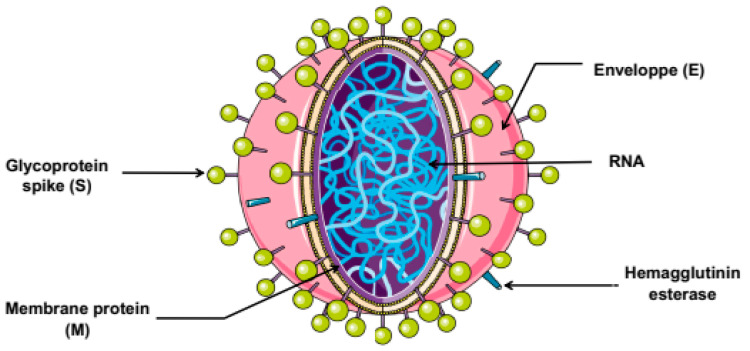
Structure of coronavirus.

**Table 1 microorganisms-08-00850-t001:** Recent tested compounds against SARS-CoV-2.

Drugs	EC_50_	CC_50_	SI	Cell Line Infected with SARS-CoV-2	References
Ivermectin	2	Not toxic	-	Vero/hSLAM cells infected with isolate Australia/VIC01/2020	[83]
Remdesivir	23.15	>100	>4.32	Vero E6 cells	[84]
Lopinavir	26.63	49.75	1.87		
Homorringtonine	2.55	ND	-		
Emetine	0.46	56.46	122.74		
Indomethacin	1	Not toxic	500	African green monkey kidney Vero E6	[85]
Nelfinavir	1.13	24.32	21.52	VeroE6 cells expressing TMPRSS2 (VeroE6/TMPRSS2)	[86]
Remdesivir	0.77	>100	>129.87	Vero E6 cells (ATCC-1586)	[87]
Chloroquine	1.13	>100	>88.50		
Ribavirin	109.50	>400	>3.65		
Penciclovir	95.96	>400	>4.17		
Nitazoxanide	2.12	>35.53	>16.76		
Nafamostat	22.50	>100	>4.44		
Favipiravir	61.88	>400	>6.46		
Hydroxychloroquine sulfate	0.72	ND	ND	C-Tan-nCoV Wuhan strain 01 (clinical isolate)	[88]
Chloroquine phosphate	5.47	ND	ND		
Hydroxychloroquine	4.06–17.31	249.50	14.41–61.45	Vero E6 cells (ATCC-1586)	[89]
Chloroquine	2.71–7.36	273.20	37.12–100.81		
Hydroxy-chloroquine	4.17	>40	>10	VeroE6 (ATCC CRL-1586)	[90]
Azithromycine	2.12	>40	>19		
Alprostadil	5.39	>40	>4.7		
Opipramol dihydrochloride	5.05	>40	>7.9		
Quinidine hydrochloride monohydrate	5.11	>40	>7.8	VeroE6 (ATCC CRL-1586)	[90]
Chlorpromazine Hydrochloride	4.03	11.88	2.94	Vero E6 cells (ATCC# CRL 1586)	[91]
Amodiaquin Dihydrochloride Dihydrate	4.94	34.42	6.97		
Imatinib Mesylate	5.32	>30.86	>5.80		
Fluspirilene	5.32	30.33	5.71		
Amodiaquin Hydrochloride	5.64	>38.63	>6.84		
Triparanol	6.41	21.21	3.31		
Clomipramine Hydrochloride	7.59	>29.68	>3.91		
Thiethylperazine Maleate	8.02	18.37	2.29		
Mefloquine Hydrochloride	8.06	18.53	2.30		
Fluphenazine Dihydrochloride	8.98	20.02	2.23		
Tamoxifen Citrate	8.98	37.96	4.23		
Promethazine Hydrochloride	10.44	>42.59	>4.08		
Hydroxychloroquine Sulfate	11.17	>50	>4.48		
Toremifene Citrate	11.30	20.51	1.81		
Terconazole Vetranal	16.14	41.46	2.57		
Benztropine Mesylate	17.79	>50	>2.81		
Chloroquine Phosphate	46.80	>50	>1.07		

IC_50_: half maximal inhibitory concentration in µM; CC_50_: half maximal cytotoxic Concentration in µM. SI: Selectivity index (CC_50_/IC_50_); ND: Not determined.

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
