# Peer review of "From Wuhan to COVID-19 Pandemic: An Up-to-Date Review of Its Pathogenesis, Potential Therapeutics, and Recent Advances"

_microorganisms, 2020, doi:10.3390/microorganisms8060850_

Round 1

Reviewer 1 Report

The topic of this review is interesting, particularly in this pandemic. Some points are given below to improve the quality of the manuscript. 

1-Title:  "a review of the novel coronavirus (SARS-CoV-2)"_ A review of what? Please, be specific. 

Another proposal for a title: From Wuhan to COVID-19 pandemic: An up-to-date review of its pathogenesis, potential therapeutics, and Recent Advances.

2-Some latest potential findings should be added. Check these papers and add text to the corresponding paragraphs. 

https://www.cell.com/cell/fulltext/S0092-8674(20)30483-9

https://www.nature.com/articles/s41591-020-0868-6

https://www.nature.com/articles/s41586-020-2179-y

https://www.sciencedirect.com/science/article/pii/S1931312820302365?via%3Dihub

3-The manuscript contains several long sentences. Please, avoid that. 

4-Figure 1: do you have permission to reuse this picture?  A citation of the source sometimes is not enough if the reference is not open access. Please, verify if it is under CC BY 4.0 license before any use. You can use the SMART database for nice free-to-use figures. Check here: https://smart.servier.com/

5-Section 4: what do you mean by other symptoms of COVID-19? I don't see any previous chapters or paragraphs on this in the manuscript. Check/correct. 

6-Cell entry and pathogenicity section: I don't see any text on the host intracellular events after viral entry. The endocytosis pathways should be added along with the key targetable signaling mediators. The target of chloroquine, for example, is in this route. The PICALM signaling should be added to this section.

Check here:  https://www.nature.com/articles/nrd.2015.37.pdf

https://www.nature.com/articles/s41565-020-0674-9

7-The danger during COVID-19 is not the virus itself but the host immune response that can be exaggerated. This is known as the cytokine storm and is potentially targetable for therapy. Tocilizumab, for example, has shown interesting efficacy. Add some text on this in the manuscript. Check here for information: https://www.nature.com/articles/s41577-020-0308-3

Congrats on the nicely written manuscript.

Author Response

Response to the Reviewers

Manuscript ID: microorganisms-801541

Response to the Reviewer 1

Dear Reviewer 1,

Thank you for the valuable comments and interesting remarks. In the following, we provide our list of changes according to your suggestions.

Thank you for your help and valuable time

Sincerely yours

Reviewer 1 report

The topic of this review is interesting, particularly in this pandemic. Some points are given below to improve the quality of the manuscript. 

Point 1. Title:  "a review of the novel coronavirus (SARS-CoV-2)"_ A review of what? Please, be specific. Another proposal for a title: From Wuhan to COVID-19 pandemic: An up-to-date review of its pathogenesis, potential therapeutics, and Recent Advances.

Authors response. We agree with reviewer 1, the proposed title is more concise and informative. (Please see page 1 lines 2-4)

Point 2. Some latest potential findings should be added. Check these papers and add text to the corresponding paragraphs. 

https://www.cell.com/cell/fulltext/S0092-8674(20)30483-9

https://www.nature.com/articles/s41591-020-0868-6

https://www.nature.com/articles/s41586-020-2179-y

https://www.sciencedirect.com/science/article/pii/S1931312820302365?via%3Dihub

Authors response. The proposed articles have been added to the corresponding paragraphs in the corrected manuscript.

https://www.cell.com/cell/fulltext/S0092-8674(20)30483-9  è (Please see page 10, lines 376-380)

https://www.nature.com/articles/s41591-020-0868-6  è (Please see page 4, line 175-177)

https://www.nature.com/articles/s41586-020-2179-y   è (Please see page 4, lines 169-170)

https://www.sciencedirect.com/science/article/pii/S1931312820302365?via%3Dihub è (Please see page 10, lines 384-387)

Point 3. The manuscript contains several long sentences. Please, avoid that. 

Authors response. This item has been resolved, and English expressions have been improved.

Point 4. Figure 1: do you have permission to reuse this picture?  A citation of the source sometimes is not enough if the reference is not open access. Please, verify if it is under CC BY 4.0 license before any use. You can use the SMART database for nice free-to-use figures. Check here: https://smart.servier.com/

Authors response. The figure 1 has been redrawn as requested. (Please see page 4)

Point 5. Section 4: what do you mean by other symptoms of COVID-19? I don't see any previous chapters or paragraphs on this in the manuscript. Check/correct. 

Authors response. This item has been corrected in the revised manuscript. (Please see page 5, line 190)

Point 6. Cell entry and pathogenicity section: I don't see any text on the host intracellular events after viral entry. The endocytosis pathways should be added along with the key targetable signaling mediators. The target of chloroquine, for example, is in this route. The PICALM signaling should be added to this section.

Check here:  https://www.nature.com/articles/nrd.2015.37.pdf

https://www.nature.com/articles/s41565-020-0674-9

Authors response.  we agree with reviewer 1, the endocytosis pathways have been added and discussed. (Please see page 4, lines 181-188)

Point 7. The danger during COVID-19 is not the virus itself but the host immune response that can be exaggerated. This is known as the cytokine storm and is potentially targetable for therapy. Tocilizumab, for example, has shown interesting efficacy. Add some text on this in the manuscript. Check here for information: https://www.nature.com/articles/s41577-020-0308-3

Congrats on the nicely written manuscript.

Authors response.

Cytokine storm data has been added to the corrected manuscript, and Tocilizumab has been discussed. (Please see page 10, lines 384-388)

Reviewer 2 Report

In this review, the authors summarized the most updated information on COVID-19 induced by SARS-CoV-2 infection. Considering the urgent nature of ongoing COVID-19 pandemic, this review seems to be very timely and will be able to serve as good resources for those who wish to obtain trustworthy information in the middle of pandemic-induced chaos. Here are some suggestions for authors to improve the quality of this paper.

  1. Overall, there are so many grammatic errors and unnatural expressions throughout the entire manuscript. I highly recommend getting a help from a native speaker for corrections. Here are some sentences or words which need more linguistic attention for clear expression.
  • “haven” at line 19
  • “over 2 878 196 confirmed 39 cases and 198 668 deaths” at line 38
  • “discovery, alternative therapies and strategies of disease control and prophylaxis as response to the COVID-19 outbreak” at line 45-> grammatic error
  • “epithelial cells more than upper airways” at line 72 -> grammatic error
  • “couldn’t be” at line 119 -> “could not be”. No abbreviation in the formal expression.
  • Hospital environment and desinfection” at line 168, -> no capital letter and misspelling.
  • “to up than” at line 173, grammatic error
  • “are ongoing” at line 203, grammatic error
  • “Table 1. Up-to-date efficient tested compounds against SARS-CoV-2.” At line 205 grammatic error
  • “it is the induce of altered ACE2 glycosylation” at line 234, grammatic error
  • “to the stop of S- protein binding” at line 234, grammatic error
  • according the” at line 240, grammatic error
  • “Covid-19 published” at line 241, “COVID-19 published”
  • “with 93%/24h” at line 245, unclear meaning
  • “develop vaccines regarding the virus target”, unclear meaning
  • “However, vs.” at line 280
  •  

  1. Description of each structural viral proteins at the “3.1. Description” section need to be combined as one paragraph. Explanation about non-structural viral protein needs to be added.
  2. A schematic picture of a virus particle in figure 1 needs to be redrawn by authors. The watermark inside the image seems to be too obvious and looks very unprofessional.
  3. The following sentence at line 111 seems to be out of context. “despite sequences diversity compared to SARS-CoV, they have demonstrated that SARS-CoV-2 causes health damage through the S-protein–ACE2 binding pathway [24].”
  4. The following sentence at line 197 also seems to be out of context. “Effectively, remdesivir and ribavirin were tested also against the novel virus. However, the efficacy pattern was different.”
  5. What does “VHH” at line 266 stand for?
  6. The meaning of “virus antibody with their clinical patient neutralization, authors” at line 270 and “at a class prediction of 100% showing” at line 271 are not clear.

Author Response

Response to the Reviewers

Manuscript ID: microorganisms-801541

Response to the Reviewer 2

Dear Reviewer 2,

Thank you for the valuable comments and interesting remarks. In the following, we provide our list of changes according to your suggestions.

Thank you for your help and valuable time

Sincerely yours

Reviewer 2 report

In this review, the authors summarized the most updated information on COVID-19 induced by SARS-CoV-2 infection. Considering the urgent nature of ongoing COVID-19 pandemic, this review seems to be very timely and will be able to serve as good resources for those who wish to obtain trustworthy information in the middle of pandemic-induced chaos. Here are some suggestions for authors to improve the quality of this paper.

Point 1.

Overall, there are so many grammatic errors and unnatural expressions throughout the entire manuscript. I highly recommend getting a help from a native speaker for corrections. Here are some sentences or words which need more linguistic attention for clear expression.

  • haven” at line 19
  • “over 2 878 196 confirmed 39 cases and 198 668 deaths” at line 38
  • “discovery, alternative therapies and strategies of disease control and prophylaxis as response to the COVID-19 outbreak” at line 45-> grammatic error
  • “epithelial cells more than upper airways” at line 72 -> grammatic error
  • “couldn’t be” at line 119 -> “could not be”. No abbreviation in the formal expression.
  • Hospital environment and desinfection” at line 168, -> no capital letter and misspelling.
  • “to up than” at line 173, grammatic error
  • “are ongoing” at line 203, grammatic error
  • “Table 1Up-to-date efficienttested compounds against SARS-CoV-2.” At line 205 grammatic error
  • “it is the induce of altered ACE2 glycosylation” at line 234, grammatic error
  • “to the stopof S- protein binding” at line 234, grammatic error
  • accordingthe” at line 240, grammatic error
  • “Covid-19 published” at line 241, “COVID-19 published”
  • “with 93%/24h” at line 245, unclear meaning
  • “develop vaccines regarding the virus target”, unclear meaning
  • “However, vs.” at line 280

Authors response.

The grammatic and linguistic errors of the manuscript have been revised. Grammatic errors have been corrected and unclear expression have been improved and reformulated. If we need more English corrections, we are at your disposal for further recommendations.  

  • haven” at line 19 è This item has been corrected (Please see page 1 line 22)
  • “over 2 878 196 confirmed 39 cases and 198 668 deaths” at line 38 è This sentence has been corrected (Please see page 1 line 43)
  • “discovery, alternative therapies and strategies of disease control and prophylaxis as response to the COVID-19 outbreak” at line 45-> grammatic errorè This sentence has been corrected and reformulated (Please see page 2 lines 51-56)
  • “epithelial cells more than upper airways” at line 72 -> grammatic errorè This item has been corrected (Please see page 2 line 81)
  • “couldn’t be” at line 119 -> “could not be”. No abbreviation in the formal expression è This item has been corrected
  • Hospital environment and desinfection” at line 168, -> no capital letter and misspelling è This item has been corrected (Please see page 6 line 246)
  • “to up than” at line 173, grammatic error è This item has been corrected (Please see page 6 line 251)
  • “are ongoing” at line 203, grammatic error è This item has been corrected (Please see page 6 line 287)
  • “Table 1Up-to-date efficienttested compounds against SARS-CoV-2.” At line 205 grammatic error (Please see page 7 line 287)
  • “it is the induce of altered ACE2 glycosylation” at line 234, grammatic error è The title of table 1 has been modified (Please see page 3 line 95)
  • “to the stopof S- protein binding” at line 234, grammatic errorè This item has been corrected (Please see page 9 line 318)
  • accordingthe” at line 240, grammatic errorè This item has been corrected
  • “Covid-19 published” at line 241, “COVID-19 published”è This item has been corrected
  • “with 93%/24h” at line 245, unclear meaningè This sentence has been corrected in the revised manuscript it means 93% in 24 hours. (Please see page 9 line 333)
  • “develop vaccines regarding the virus target”, unclear meaningè This sentence has been reformulated in the revised version, we mean that vaccine development is based also on the virus target such as spike proteins, enzymes etc (Please see page 9 line 348-350)
  • “However, vs.” at line 280è This item has been corrected to VS as abbreviation of virtual screening.

Point 2.

Description of each structural viral proteins at the “3.1. Description” section need to be combined as one paragraph. Explanation about non-structural viral protein needs to be added.

Authors response.

We agree with the reviewer, description of structural viral proteins need to be combined as one paragraph. We have combined this into one coherent paragraph. In addition, we have added more details of the nonstructural proteins (nsps) and their involvement in the viral life cycle. Also, interaction between the most studied nsp complex has been discussed. Recent investigations of these proteins related to COVID-19 has been reported in the revised manuscript. (Please see pages 2 and 3, lines 86-136)

Point 3.

A schematic picture of a virus particle in figure 1 needs to be redrawn by authors. The watermark inside the image seems to be too obvious and looks very unprofessional.

Authors response.

The figure 1 has been redrawn as requested. (Please see page 4)

Point 4.

The following sentence at line 111 seems to be out of context. “despite sequences diversity compared to SARS-CoV, they have demonstrated that SARS-CoV-2 causes health damage through the S-protein–ACE2 binding pathway [24].”

Authors response.

This sentence has been removed.

Point 5.

The following sentence at line 197 also seems to be out of context. “Effectively, remdesivir and ribavirin were tested also against the novel virus. However, the efficacy pattern was different.”

Authors response.

We agree with reviewer, this sentence is misplaced. It has been removed from the revised manuscript.

Point 6.

What does “VHH” at line 266 stand for?

Authors response.

The VHH refers to Camelid antibodies that have only two heavy chains connected by disulphide bonds in the hinge region. These heavy chain antibodies take VHH as a nomination and not abbreviation. (Please see page 9 line 353-357)

Point 7.

The meaning of “virus antibody with their clinical patient neutralization, authors” at line 270 and “at a class prediction of 100% showing” at line 271 are not clear.

Authors response.

This sentence has been reformulated in the corrected manuscript; we mean that authors have screened a list of antibodies sequences in a hospital. The information contains antibody-antigen sequences of various viruses such as SARS, Ebola and Influenza along with their IC50 values of neutralization panel and data about the patient response. This was conducted in order to have an idea about their inhibitory effects and efficacy, then to predict the activity against COVID-19 using a machine learning model. The results have shown that statistically, the prediction class is significant 100%. (Please see page 9 line 359-366)

Round 2

Reviewer 2 Report

No further comments.